

# Spatial variability of some soil properties in west coastal area of India having oil palm (*Elaeis guineensis* Jacq.) plantations

**Sanjib K. Behera[1] K. Suresh[1], B. N. Rao[1], Ravi K. Mathur[1], Arvind K. Shukla[2], K. Manorama[1], K. Ramachandrudu[1], P. Harinarayana[1]**

[1]ICAR-Indian Institute of Oil Palm Research, Pedavegi, West Godavari, Andhra Pradesh 534450, India

[2]ICAR-Indian Institute of Soil Science, Nabibagh, Berasia Road, Bhopal, Madhya Pradesh 462038, India

*Correspondence to:* Sanjib K. Behera (sanjibkumarbehera123@gmail.com)

**Running title:** Soil property distribution in oil palm plantations of coastal India

**Abstract.** Mapping spatial variability of soil properties is the key to efficient soil resource management for sustainable crop yield in coastal areas. Therefore, the present study was conducted to assess the spatial variability of soil properties like – acidity (pH), salinity (Electrical Conductivity (EC)), organic carbon, available K, available P, exchangeable $Ca^{2+}$, exchangeable $Mg^{2+}$, available S and hot water soluble B in surface (0-20 cm) and subsurface (20-40 cm) soil layers of oil palm plantations in south Goa and north Goa districts of Goa situated in west coastal area of India. A total of 128 soil samples were collected from 64 oil palm plantations of Goa located at an approximate interval of 5-7 km and analyzed. Soil was acidic to neutral in reaction. Other soil properties varied widely in both the soil layers. Correlations between soil pH and exchangeable $Ca^{2+}$, between soil EC and available K, between available P



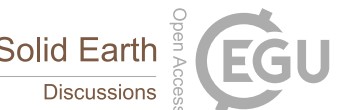

and available S and between exchangeable $Ca^{2+}$ and exchangeable $Mg^{2+}$ in both the soil layers
were found to be positive and significant (P = 0.01). Geostatistical analysis revealed different
spatial distribution pattern for the measured soil properties. Best fit models of measured soil
properties were spherical, linear, exponential, circular and Gaussian with weak to strong spatial
dependency. The results revealed that site-specific fertilizer management options needed to be
adopted in the oil palm plantations of the study area owing to variability in soil properties.
*Keywords:*  Soil management, Spatial distribution, Precision agriculture, Soil fertility, Coastal
zone
**1 Introduction**
Soil is the key part of the earth system which controls hydrological, biological, and
geochemical cycles and it offers goods, resources and services to mankind (Keesstra et al., 2012;
Smith et al., 2015; Decock et al., 2015; Brevik et al., 2015; Berendse et al., 2015). Un-
sustainable soil management practices lead to soil degradation, which is a worldwide topic,
mainly because of loss of soil organic matter (SOM), soil erosion, changes in soil structure,
degradation of the biota in the soils and soil chemical degradation (Cerda et al., 2009; Mupenzi
et al., 2011; Novara et al., 2013; Mukherjee et al., 2014 Lieskovsky and Kenderessy, 2014;
Stanchi et al., 2015; Seutloali and Beckedahl, 2015; Novara et al., 2015;). Soil degradation along
with natural processes results in degradation of coastal areas, which covers more than 10% of the
earth surface area with 35, 6000 and 7517 km coast line in world and India respectively
(Misdorp, 1990; Sanil Kumar et al., 2006). Therefore, there is a need to describe and characterize
these areas for adoption of effective land use practices including application of agri-inputs
(Arakel et al., 1993; Guneroglu et al., 2015).



Geographical distribution maps of soil properties, obtained from soil surveys, help in
correct management of soil nutrients (Brevik et al., 2016). These maps are required to understand
the patterns and processes of soil spatial variability, which is the combined effect of soil
physical, chemical and biological processes operating at different spatio-temporal scales
combined with anthropogenic activities (Goovaerts, 1998). The distribution maps are prepared
by analysing spatial distribution pattern of soil properties. Geostatistical tools are useful in
preparation of the maps based on limited number of samples collected from agricultural
landscapes. These tools predict the values at un-sampled locations by spatial correlation and
reducing variance of estimation error and investigation costs (Saito et al., 2005; Pereira et al.,
2015). Spatial variability of soil properties is assessed effectively by geostatistical methods
(Mueller et al., 2003) for site-specific management of nutrients through variable rate fertilizer
application to avoid over and under application of nutrients.  Li et al., 2011, Behera and Shukla,
2014 and Behera and Shukla, 2015 have reported different spatial variability pattern of soil
properties and soil nutrients in eroded areas of south China and some cultivated acid soils of
India. Information regarding variability of soil properties in soil profile is helpful to assess the
contribution of sub-surface soil layers to crop nutrition and potential capacity of the soil to
supply nutrients during crop growth. It also helps in understanding the effect of different
management practices, under a given cropping system, on the downward movement as well as
recycling of nutrients to the surface layers (Behera and Shukla, 2013, Parras-Alcantara et al.,

64   2015).

Oil palm (*Elaeis guineensis* Jacq.) is a high oil yielding crop (Lamade et al., 2015). On
average, it produces ten times more oil than any leading oil seed crop from a hectare of land and
some efficient farmers get as high as eight tonnes of oil yield per hectare. World-wide, oil palm



produces 32% oils and fats output from 5.5% land use for cultivation (Palm Oil Research, 2016).
Indonesia and Malaysia are the leading producer of oil palm. According to Rethinam et al.
(2012), oil palm can be cultivated as irrigated crop in 1.93 million ha area in 18 states of India.
At present, oil palm is being grown in an area of about 2, 68, 000 ha covering twelve states of the
country, having different soil types, with productivity levels reaching as high as 30-35 Mg fresh
fruit bunches (FFB) ha$^{-1}$ year$^{-1}$ (Kalidas et al., 2015).

74       Rationale use of fertilizer results in environmentally sustainable and economically viable

oil palm yield (Goh et al., 2003). Oil palm uses about 162, 30, 217, 38 and 36 kg of N, P, K, Mg
and Ca ha$^{-1}$ year$^{-1}$ respectively, to produce 2.5 Mg of oil ha$^{-1}$ year$^{-1}$ (Mengel and Kirkby, 1987).
Considering oil to bunch ratio of 1:4, 2.5 Mg oil ha$^{-1}$ is equivalent to 10 Mg FFB ha$^{-1}$ year$^{-1}$, but
average FFB yield in well-managed plantations is much higher (Narsimha Rao et al., 2014).
Nutrient content in 1 Mg of FFB obtained from Dura palms is 2.94, 0.44, 3.71, 0.77, 0.81 kg of
N, P, K, Mg and Ca, respectively, whereas, Mn, Fe, B, Cu and Zn content per 1 Mg of FFB is
1.51, 2.47, 2.15, 4.76 and 4.93 g, respectively of Mn, Fe, B, Cu and Zn (Ng and Thamboo,
1967). According to Narsimha Rao et al. (2014), nutritional problems like N/K imbalance, K
deficiency, Mg deficiency and B deficiency affect oil palm production in oil palm plantations of
India. Calibrated soil and leaf analysis helps in effective fertilizer recommendations in most of
the crops (Smith and Loneragan, 1997; McLaughlin et al., 1999). In oil palm, leaf nutrient
analysis is commonly used for estimating fertilizer requirement (Fairhurst and Mutert 1999;
Corley and Tinker, 2003). The relationship between leaf analysis and palm productivity is
generally evident, and an assessment of fertilizer needs can be based on such an analysis.
However for a cost-effective approach, leaf analysis has to be integrated with soil analysis (Goh



et al., 2009). It is therefore pertinent to assess soil and leaf nutrient status for effective and
sustainable fertilizer management programme in oil palm.
The nutrient management recommendations in oil palm plantations in India in general
and oil palm plantations in the area under study are generic ones. Prasad et al. (2013) reported
wide range in quantity of fertilizer applied indicating that oil palms were either under-fertilized
or over-fertilized. Also, low cost and easy availability of some fertilizers have encouraged
farmers to make excessive applications with the belief that high yields would be ensured.
However, this management adversely affects soil fertility, productivity, fruit quality and ground
water quality. It is therefore pertinent for the farmers to economize on fertilizer adopting a
strategy for site-specific and/or area-specific management based on spatial variability of soil
properties to make oil palm production environmentally sustainable and economically viable.
Spatial variability of soil properties in oil palm plantations have to be carefully evaluated to
carryout sustainable soil management practices. Thus, the present study was carried out in soils
of oil palm plantations of Goa state of India with the following objectives, (i) to estimate the
spatial variability of some soil properties through semivariogram analysis, (ii) to develop spatial
maps for soil properties using the parameters of the best fitted semivariogram model and
interpolation by ordinary kriging technique and (iii) to assess the relationship among the
estimated soil properties.
**2 Material and methods**
**2.1 Study site**
A survey was carried out in Goa state of India during 2012-13 to find out soil and plant
nutritional status in randomly selected 64 tenera oil palm plantations (with 5 to 21 years of age)



(Figure 1). Oil palm is cultivated in an area of approximately 1000 ha which is 1% of agricultural

land in the state. The sampling area lies between $15^{°}$ 6.8 96 N to $15^{°}$ 41.7 26 N latitudes and

$74^{°}$ 76 60 to $73^{°}$ 56 78 E longitudes with altitude ranging from 4 to 90 meter above sea level.

The climate of the area is tropical monsoon type. Hot and humid climate prevails for most of the

year. Annual mean rainfall (average of 30 years) is 2926 mm, concentrated from early June to

late September. On average, May is the warmest month, with temperature peaks over 35 °C

(during 24 h) and relative humidity of 70%. Goa experiences short winter seasons between mid-

December and February and these months are marked by mean night temperature of

approximately 21 °C and mean day temperature of around 28 °C with relative humidity of

65%. According to Bhattacharyya et al., (2013), the main soils in the study area are Inceptisols

(26, 000 ha), Ultisols (4, 000 ha), Entisols (3,000 ha) and Alfisols (3, 000 ha) (classified as in

Soil Survey Staff, 2014), sandy loam to silty loam texture, developed from granite, granite-

gneiss, quartzite/schistose and basalt.

**2.2 Soil sampling, processing and analysis**

A total of 128 soil samples i.e. 64 from 0-20 cm (surface) and 64 from 20-40 cm (sub-

surface) depths were collected at random points inside 3-m radius from the palm during the

survey to assess soil properties of oil palm plantations at an approximate interval of 5 to 7 km.

All the samples were collected with a hand auger. The latitude, longitude, and elevation at each

sampling point were recorded using a hand held global positioning system (GPS). The soil

samples were dried at room temperature ($25 \pm 3$ $^{0}$C). Roots and debrises were removed from the

samples by hand. Samples were processed following standard procedures. The processed soil

samples were tested for acidity (pH), salinity (EC), organic carbon (OC) content, available K

($NH_4OAc$-K), available P (Bray's P-1) (Bray's-P), exchangeable $Ca^{2+}$, exchangeable $Mg^{2+}$,



available S (CaCl$_2$-S) and hot water extractable B (HWB). Determination of soil pH and EC
(1:2 soil water ratio (w/v) suspension) were carried out using pH-meter and conductivity meter
(Jackson, 1973). Walkley-Black method (Walkley and Black, 1934) was followed for assessing
soil OC content. NH$_4$OAc-K was estimated after extracting soil samples with neutral 1 N
ammonium acetate solution (Hanway and Heidel, 1952) followed by flame photometry
estimation.  Available P was extracted using Bray's P-1 reagent (Bray and Kurtz, 1945) and
estimated through spectrophotometry. Ca$^{2+}$ and Mg$^{2+}$ were extracted using    neutral normal
ammonium acetate solution (Jones, 1998) and estimated through atomic absorption spectrometry.
Available S was estimated by the turbidity method (Williams and Steinbergs, 1969). HWB
content was estimated through Azomethine-H reagent (Gupta, 1967) using spectrophotometry.

**2.3 Statistical and geostatistical analysis**

146        The descriptive statistics like minimum, maximum, mean, standard deviation (SD),

coefficient of variation (CV), and skewness for soil properties were computed using the SAS 9.2
software pack (SAS, 2011). Relationship among the estimated soil properties were established
using Pearson's correlation coefficient analysis at p   0.05 and p   0.01.

150        ArcMap 10.1 (ESRI, 2012) was used to analyze the spatial structure of soil properties.

Before using geostatistics, normality of data distribution were checked by Shapiro-Wilk test at 5%
(Shapiro and Wilk, 1965). Soil properties like pH and OC content in both the soil layers and CaCl$_2$-
S content in subsurface soil layers exhibited normal distribution (Table 1). While, data
transformation to normal distribution was carried out for rest of the soil properties. Trend of the
data set was checked and removed. The semivariogram models of soil properties were derived as
described by Goovaerts (1997) and Tesfahunegn et al. (2011).



$$\gamma(h) = \frac{1}{2m(h)} \sum_{i=1}^{m(h)} [Z(X_i + h) - Z(X_i)]^2$$


(1)

Where    $(h)$ is the experimental semivariogram, $m(h)$ is number of sample value pairs,
$Z(X_i)$, $Z(X_i+h)$ are sample values at two points. Best fitted semivariogram model for each soil
property was selected by using the cross validation technique.
Semivariogram parameters like nugget/sill ratio and range were obtained for soil
properties. The nugget/sill ratio expressed in percentage was used to classify the spatial
dependence of variables (Oliver and Webster, 2014). Ratio values less than or equal to 25%,
between 25 and 75%, more than 75% were considered strongly, moderately and weakly spatially
dependent, respectively (Behera et al., 2011). Best-fit semivariograms models were selected by
cross-validation technique. Mean square error (MSE) was estimated to predict the accuracy of
models (Utset et al., 2000).

$$\mathbf{MSE} = \frac{\sum_{i=1}^{n} [z(x_i, y_i) - z * (x_i, y_i)]^2}{n}$$


(2)

Accuracies of interpolated maps were checked by the goodness-of-prediction criterium G
(Agterberg, 1984).  According to Parfitt et al. (2009), positive and negative and close to zero
values of  G indicate that the map obtained by interpolating data from the samples is more
accurate than average value of the area and the average value predicts the values at un-sampled
locations as accurately as or even better than the sampling estimates, respectively. Ordinary
kriging interpolation was carried out to develop spatial distribution maps for soil properties.
**3 Results and discussion**
**3.1 Descriptive statistics of soil properties**



The descriptive statistics revealed considerable variability of soil properties in both
surface and sub-surface soil layers of oil palm plantations (Table 1). The values of CV for soil
acidity in both the soil layers revealed their low variability (CV < 10%) (Nielsen and Bouma,
1985).   The rest of the soil properties exhibited moderate (CV 10 to 100%) variability except
salinity in surface soil layers and Bray's-P in both the soil layers, which had high (CV > 100%)
variability. Low CV values for soil acidity was due to transformed measurement of hydrogen ion
concentration.  Skewnees coefficient values of 0.18 to 3.89 for different soil proprieties revealed
that some soil properties were not normally distributed.   This variation and non-normal
distribution of soil properties in the studied areas may be due to adoption of different soil
management practices including variation in fertilizer application and other crop management
practices (Tesfahunegn et al., 2011; Srinivasarao et al., 2014; Ferreira et al., 2015).
The mean values of soil pH were acidic in both surface (5.35) and subsurface (5.28) soil
layers (Table 1). The acidic nature of soil in the studied area may be due to acidic parent material
and prevailing rainfall pattern. The values of soil EC indicate the non-saline nature of soils.  Soil
OC contents varied widely in both surface and subsurface soil layers. Principal reason for
variation in soil OC content may be due to adoption of different cultural practices including
addition of crop biomass to the soils. Surface soil layers had slightly higher OC content (mean
value 19.8 g kg$^{-1}$) than OC content in subsurface soil layers (mean value 13.2 g kg$^{-1}$). Surface
soil layers had higher $NH_4OAc$-K, Bray's-P, $CaCl_2$-S and HWB content compared to that in
subsurface soil layers (Table 1). The content of these nutrients varied greatly among the soils
because of heterogeneity in fertilizer application in the area. The mean values of exchangeable
$Ca^{2+}$ were 914 and 795 mg kg$^{-1}$ for surface and subsurface soil layers, respectively. Whereas,
surface soil layers were having 203 and 225 mg kg$^{-1}$ of mean exchangeable $Mg^{2+}$ content,



respectively. Other studies reported similar results highlighting different distribution pattern of
soil properties, primary, secondary and micronutrients under different soil-crop management
situations (Franzlubbers and Hons, 1996; Sharma et al., 2005; Behera and Shukla, 2013).
**3.2 Relationship among soil properties**
The exchangeable $Ca^{2+}$ content increased with pH (Table 2). Behera and Shukla (2015)
also recorded positive and significant relationship of soil pH and soil OC with K, exchangeable
$Ca^{2+}$ and exchangeable $Mg^{2+}$ content in some cropped acid soils of India. Soil OC content in
surface layers was positively and significantly correlated with exchangeable $Ca^{2+}$ and HWB (P
0.05). Most of the soil properties which influence nutrient storage and availability to plants are
influenced by soil organic matter (SOM) type and content (Foth and Turk, 1972). Increased soil
EC content led to higher $NH_4OAc$-K in both soil layers (P    0.01), and higher $CaCl_2$-S in surface
layer and Bray's–P in subsurface layer (P    0.05). Soil EC does not directly affect plant growth
but has been used as an indirect indicator of the amount of nutrients available for plant uptake
and salinity levels (Corwin and Lesch, 2005). EC has been used as a surrogate measure of salt
concentration, organic matter, cation-exchange capacity, soil texture, soil thickness, nutrients,
water-holding capacity, and drainage conditions. In site-specific management and high-intensity
soil surveys, EC is used to partition units of management, differentiate soil types, and predict soil
fertility and crop yields.
**3.3 Spatial structure and distribution of soil properties**
The best-fit semivariogram models and parameters of studied soil properties are given in
Table 3. The best fit models for soil properties of studied areas were spherical, linear,



exponential, circular, and Gaussian depending on soil layer and parameter. Our findings are in line with the observations made by Tesfahunegn et al. (2011).

Cross-validation technique was used to select semivariograms models for soil properties with the lowest MSE values (Table 3). Lowest MSE values indicate that kriging predictions of soil properties are closer to measured values. The accuracy of kriged interpolation maps of soil properties was also measured by the G values (Table 3) which varied from 26 (for exchangeable $Ca^{2+}$ in subsurface layer) to 76% (for HWB in subsurface layer). Positive G values for all the soil properties revealed the developed maps are more accurate than the maps generated using the average value of the area.

Soil pH in both the soil layers of oil palm plantations was having moderate spatial dependency class. Soil EC had strong and moderate spatial dependency for surface and sub-surface soil layers respectively. Soil OC content in oil palm plantations had weak spatial dependency for surface soil layers and moderate spatial dependency for sub-surface soil layers. Spatial dependency classes were weak for $NH_4OAc$-K and strong for exchangeable $Ca^{2+}$ for both the soil layers of oil palm plantations. Bray's–P and $CaCl_2$-S had weak spatial dependency for surface layers and moderate spatial dependency for sub-surface layers. Whereas, exchangeable $Mg^{2+}$ and HWB had moderate spatial dependency in surface soil layers and weak spatial dependency for sub-surface soil layers in oil palm plantations. Weak spatial dependency of soil properties like $NH_4OAc$-K, Bray's–P and OC (in surface layer) in oil palm plantations is ascribed to the anthropogenic activities like adoption of cultural practices including application of fertilizers. In these oil palm plantations, activities like application of irrigation water, weeding, basin cleaning, mulching and application of N, P, K and Mg fertilizer are carried out at regular

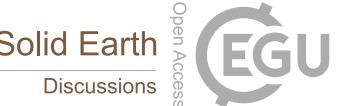

intervals. Whereas, moderate and strong spatial dependency of soil pH, EC and exchangeable
$Ca^{2+}$ is due to soil type and parent material.
According to Webster and Oliver (1990), range value is a measure of the spatial
extension within which autocorrelation exists. Spatially related samples were separated by
distances closer than range values. The range values of soil properties in studied area varied
widely (Table 3). The range values for surface soil properties were 554 to 4530 m and for
subsurface soil properties were 581 to 4530 m. Among the soil properties, higher range values
were recorded for $NH_4OAc$-K and $CaCl_2$-S for both the soil layers. The possible causes for
spatial variability of soil properties in studied areas are adoption of different soil management
practices (Bodi et al., 2013; Pereira et al., 2013; Ochoa-Cueva et al., 2015). The difference in
annual average temperature in the state of Goa was more than 12 °C, indicating temperature
could be important factor influencing soil nutrient mineralization and accumulation. Moreover,
this area is having rising slope from the coast line towards *ghats* i.e. from western side to eastern
side, which could also affect distribution of nutrients probably wash by surface runoff or
subsurface water movements.
Interpolation maps (Figure 2) of different soil properties revealed that oil palm
plantations of the area could be divided into homogenous small zones depending upon the
different nutrient ranges.  Overlying of the spatial distribution maps on map of Goa revealed that
the spatial distribution map of pH in surface soil layers revealed almost all the area having pH of
5.00 to 6.00. Low pH values occurred in north-western and south-eastern parts. In sub-surface
soil layers, low pH of 4.75 to 5.00 occurred in south-eastern part whereas relatively higher pH
prevailed in north-western part. Areas having low pH values compared to other areas may be due
acidic parent material from which the soil developed and different soil management practices.





Accordingly, different management options may be adopted in different parts of the area with
different levels of pH. Soil EC had irregular distribution pattern in surface soil layers whereas
low values of EC were recorded in north-western part. This may be due to sandy loam soil
texture and presence of low OC in north-western part. Higher EC values in other parts of
surveyed area probably due to silt loam soil texture with high water table. Higher amount of soil
OC was found to be distributed in the south-eastern parts in surface as well as sub-surface soil
layers. This may be ascribed to prevalence of higher slope and low rate of SOM mineralization
in south-eastern parts compared to other areas. Higher amount of $NH_4OAc$- K and $CaCl_2$-S was
found to be distributed in almost all parts in both the soil layers. Higher amount of Bray's-P was
found to be distributed in almost all parts in surface soil layers whereas low amount of Bray's-P
occurred in north-central and south-western parts in sub-surface soil layers. Build up of P in
surface layers may be due to continuous P addition and their fixation in soil which is acidic in
nature. Exchangeable $Ca^{2+}$ exhibited irregular distribution pattern in both the soil layers. In
surface as well as sub-surface soil layers, lower amount of Exch. Mg was found to be distributed
in southern parts as compared to that in northern parts. Similar distribution of exchangeable $Ca^{2+}$
and exchangeable $Mg^{2+}$ was recorded in these soils which corroborate our finding of significant
and positive correlation between exchangeable $Ca^{2+}$ and exchangeable $Mg^{2+}$ in both the soil
layers. Higher amount of HWB was found to be distributed in north-eastern part in surface soil
layers and in central and south-western parts in sub-surface soil layers. The different distribution
variability of the soil properties in oil palm plantations of this area is predominantly due to
climate and landscape along with farm practices including application of different quantities of
nutrients through fertilizers. The kriged distribution maps for different soil properties providing
quantitative information about  soil properties in both the soil layers is of great use for plantation



staff, farm managers, extension officers and farmers. This will help in visualizing soil fertility
status for planning appropriate strategies for efficient site specific soil nutrient management and
variable-rate fertilizer application technology. It leads for obtaining optimum output and oil palm
yield which can provide environmentally sustainable maximum return to famers with optimum
input utilization combined with best management practices (Fu et al., 2010; Behera et al., 2012).
The areas with low and medium nutrient status require more amount of fertilizer application as
compared to areas having high nutrient status. For example, exchangeable $Mg^{2+}$ status is low in
southern part of the area compared to northern part. Therefore, the requirement of Mg fertilizer
application is more in southern part compared to northern part.
**4 Conclusions**
Geostatistical analysis is the key for studying the spatial variability of soil properties for
sustainable soil resource management. The present study divulged that the measured soil
properties had large variability in spatial distribution pattern in both surface and subsurface soil
layers of oil palm plantations of the studied area. Positive and significant correlations were
recorded between soil pH and exchangeable $Ca^{2+}$, soil EC and $NH_4OAc$-K, Bray's-P and $CaCl_2$-
S and exchangeable $Ca^{2+}$ and exchangeable $Mg^{2+}$ in both the soil layers. The prediction maps
generated by geostatistical analysis are useful for site-specific soil nutrient management in oil
palm plantations of the area by delineating management zones and adoption of variable fertilizer
application strategies.
*Acknowledgements.* The authors gratefully acknowledge the help rendered by M/s Godrej
Agrovet Private Limited., India for collection of soil samples.  The authors thank Prof. Artemi





Cerda and the topical editor Prof. Paulo Pereira for their suggestions for improvement of the
manuscript.

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





**Table 1.** Soil properties of surface (0-20 cm) and sub-surface (20-40 cm) layers (n = 64 at each case).

| Variable | Soil layer | Mean ± SD | CV (%) | Minimum | Maximum | Skewness coefficient | Distribution |
|---|---|---|---|---|---|---|---|
| pH | Surface | 5.35± 0.45 | 8.64 | 4.25 | 6.77 | 0.18 | Normal |
| | Subsurface | 5.28± 0.46 | 8.63 | 4.53 | 6.52 | 0.65 | Normal |
| EC | Surface | 0.13±0.17 | 125 | 0.05 | 1.06 | 4.06 | Transformed |
| | Subsurface | 0.08±0.06 | 75.3 | 0.03 | 0.41 | 3.02 | Transformed |
| OC | Surface | 19.8±8.77 | 44.4 | 5.07 | 48.4 | 0.83 | Normal |
| | Subsurface | 13.2±7.33 | 55.5 | 1.95 | 31.2 | 0.75 | Normal |
| $NH_4OAc$-K | Surface | 270±29.9 | 88.7 | 58.1 | 1167 | 1.80 | Transformed |
| | Subsurface | 199±165 | 82.8 | 16.1 | 856 | 2.16 | Transformed |
| Bray's-P | Surface | 24.7±3.39 | 127 | 0.86 | 141 | 2.14 | Transformed |
| | Subsurface | 9.78±13.2 | 135 | 0.90 | 42.3 | 2.52 | Transformed |
| $Ca^{2+}$ | Surface | 914±588 | 64.3 | 200 | 2997 | 1.56 | Transformed |
| | Subsurface | 795±724 | 91.1 | 194 | 5177 | 3.89 | Transformed |
| $Mg^{2+}$ | Surface | 203±141 | 69.3 | 36.0 | 744 | 1.75 | Transformed |
| | Subsurface | 225±156 | 69.4 | 24.0 | 720 | 1.27 | Transformed |
| $CaCl_2$-S | Surface | 23.2±16.4 | 70.7 | 3.00 | 87.7 | 1.60 | Transformed |
| | Subsurface | 16.3±10.1 | 62.0 | 1.50 | 43.5 | 0.93 | Normal |
| HWB | Surface | 0.70±0.38 | 54.7 | 0.09 | 2.10 | 1.43 | Transformed |
| | Subsurface | 0.64±0.44 | 68.6 | 0.04 | 2.56 | 1.70 | Transformed |

SD-standard deviation; CV-coefficient of variation; EC-electrical conductivity, dS m$^{-1}$; OC-organic carbon, g kg$^{-1}$; K, mg kg$^{-1}$; P, mg kg$^{-1}$; exchangeable Ca$^{2+}$, mg kg$^{-1}$; exchangeable Mg$^{2+}$, mg kg$^{-1}$; S, mg kg$^{-1}$;HWB, hot water soluble B, mg kg$^{-1}$.





**Table 2.** Pearson's correlation coefficients between soil properties at the surface (0-20 cm) and subsurface (20-40 cm) layers. Only significant coefficients are shown (*, p ≤ 0.05; **, p ≤ 0.01) (n=64).

| Layer | | pH | EC | OC | P | Ca$^{2+}$ |
|---|---|---|---|---|---|---|
| Surface | K | | 0.45** | | | |
| | P | | | | | |
| | Ca$^{2+}$ | 0.67** | | 0.26* | | |
| | Mg$^{2+}$ | | | | | 0.37** |
| | S | | 0.31* | | 0.44** | |
| | HWB | | | 0.30* | | |
| Sub-surface | K | | 0.48** | | | |
| | P | | 0.32* | | | |
| | Ca$^{2+}$ | 0.42** | | | | |
| | Mg$^{2+}$ | | | | | 0.33** |
| | S | | | | 0.36** | |

EC-electrical conductivity, dS m$^{-1}$; OC-organic carbon, g kg$^{-1}$; K, mg kg$^{-1}$; P, mg kg$^{-1}$; exchangeable Ca$^{2+}$, mg kg$^{-1}$; exchangeable Mg$^{2+}$, mg kg$^{-1}$; S, mg kg$^{-1}$;HWB, hot water soluble B, mg kg$^{-1}$.



**Table 3.** Semivariogram parameters of soil properties of studied areas.

| Variable | Soil layer | Model | Nugget | Sill | Nugget: Sill ratio | Spatial class | Range (m) | MSE | G (%) |
|---|---|---|---|---|---|---|---|---|---|
| pH | Surface | Spherical | 0.098 | 0.130 | 0.715 | Moderate | 1416 | 0.754 | 62 |
| | Subsurface | Spherical | 0.110 | 0.160 | 0.687 | Moderate | 1468 | 0.681 | 58 |
| EC | Surface | Spherical* | 0.001 | 0.004 | 0.025 | Strong | 554 | 0.0003 | 55 |
| | Subsurface | Linear* | 0.003 | 0.004 | 0.750 | Moderate | 2186 | 0.0002 | 51 |
| OC | Surface | Exponential | 54.10 | 67.70 | 0.797 | Weak | 1131 | 2.31 | 48 |
| | Subsurface | Circular | 20.80 | 51.10 | 0.407 | Moderate | 581 | 3.12 | 56 |
| NH$_4$OAc-K | Surface | Spherical* | 36371 | 40122 | 0.906 | Weak | 4530 | 28.31 | 65 |
| | Subsurface | Linear* | 21523 | 22506 | 0.956 | Weak | 4530 | 30.01 | 60 |
| Bray's-P | Surface | Gaussian* | 875.0 | 940.0 | 0.930 | Weak | 1996 | 40.02 | 53 |
| | Subsurface | Gaussian* | 97.60 | 149.9 | 0.651 | Moderate | 770 | 39.58 | 50 |
| Ca$^{2+}$ | Surface | Linear* | 0.000 | 263780 | 0.000 | Strong | 1585 | 221.01 | 33 |
| | Subsurface | Exponential* | 0.000 | 330416 | 0.000 | Strong | 581 | 198.65 | 26 |
| Mg$^{2+}$ | Surface | Gaussian* | 11244 | 21059 | 0.533 | Moderate | 885 | 89.56 | 50 |
| | Subsurface | Exponential* | 19839 | 20685 | 0.959 | Weak | 1114 | 70.04 | 53 |
| CaCl$_2$-S | Surface | Linear* | 234.0 | 245.0 | 0.955 | Weak | 4530 | 0.067 | 45 |
| | Subsurface | Gaussian | 62.10 | 93.20 | 0.666 | Moderate | 4530 | 0.071 | 42 |
| HWB | Surface | Gaussian* | 0.046 | 0.073 | 0.630 | Moderate | 1424 | 0.023 | 71 |
| | Subsurface | Linear* | 0.111 | 0.147 | 0.755 | Weak | 1148 | 0.018 | 76 |

*Transformation for normal distribution.
EC-electrical conductivity, dS m$^{-1}$; OC-organic carbon, g kg$^{-1}$; K, mg kg$^{-1}$; P, mg kg$^{-1}$; exchangeable Ca$^{2+}$, mg kg$^{-1}$; exchangeable Mg$^{2+}$, mg kg$^{-1}$; S, mg kg$^{-1}$;HWB, hot water soluble B, mg kg$^{-1}$;  MSE-mean square error; G-goodness-of-prediction criterium.





**Figure 1.** Spatial distribution of sampling points in Goa state (western India)

**Figure 2.** Kriged interpolation maps of soil properties in surface (0-20 cm) and subsurface (20-40 cm) soil layers



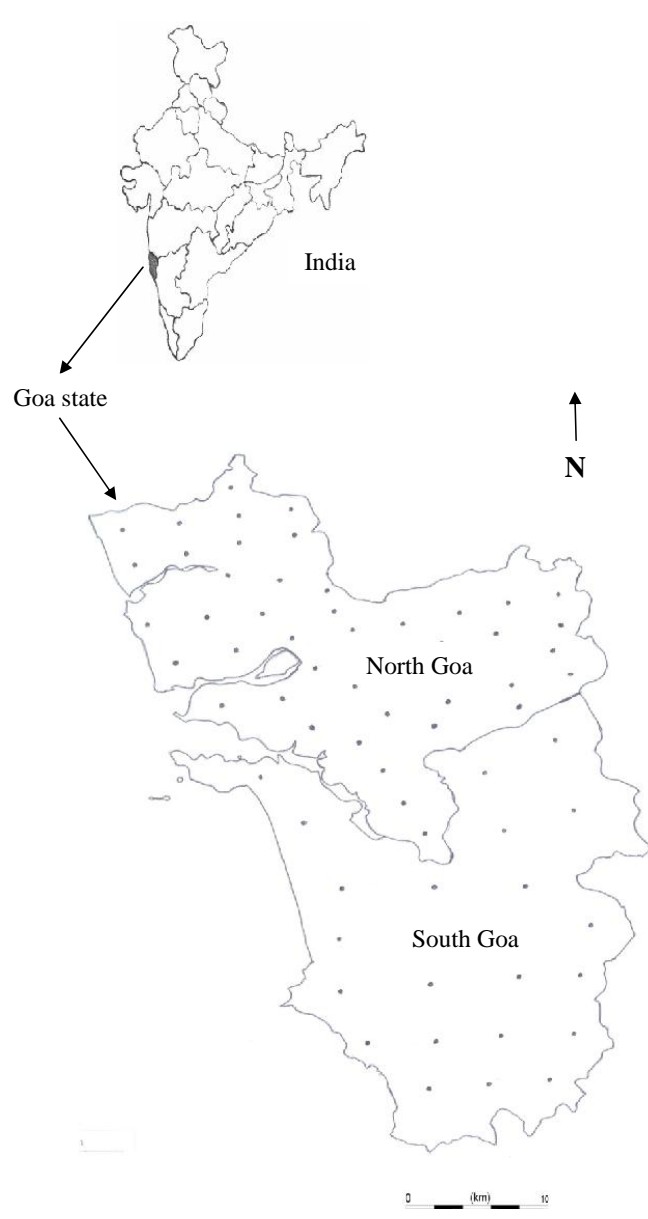

**Figure 1.** Spatial distribution of sampling points in Goa state (western India)

pH





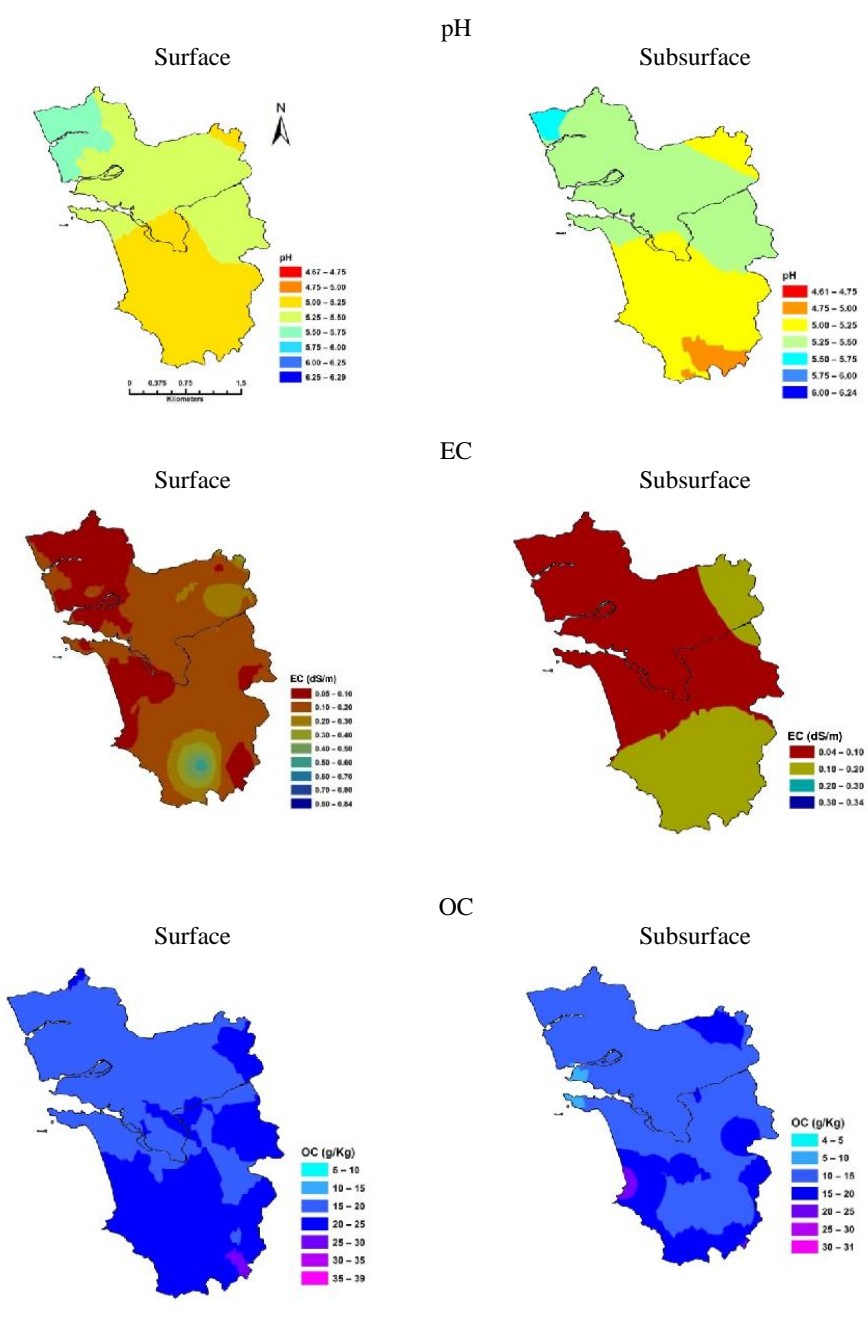



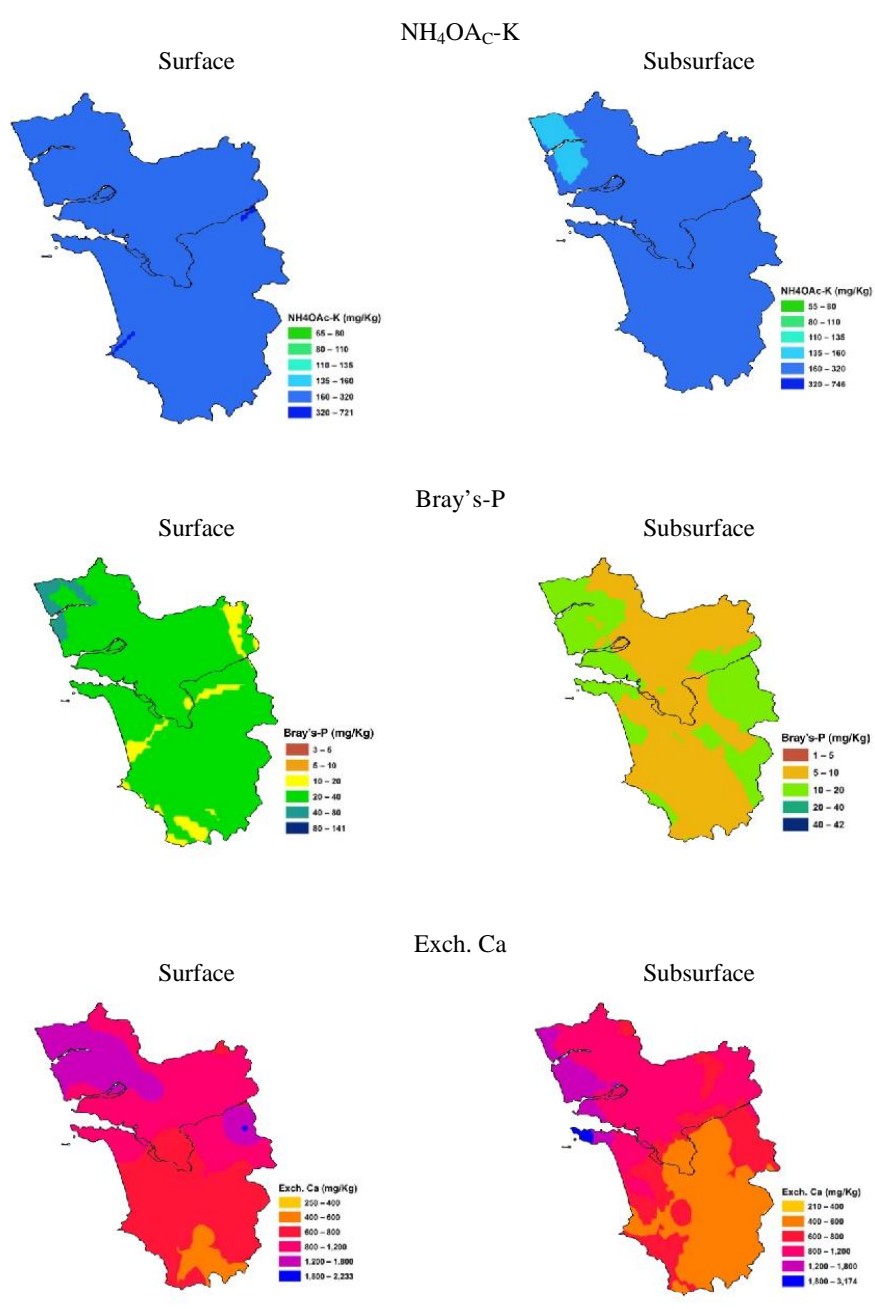

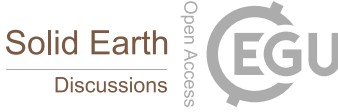



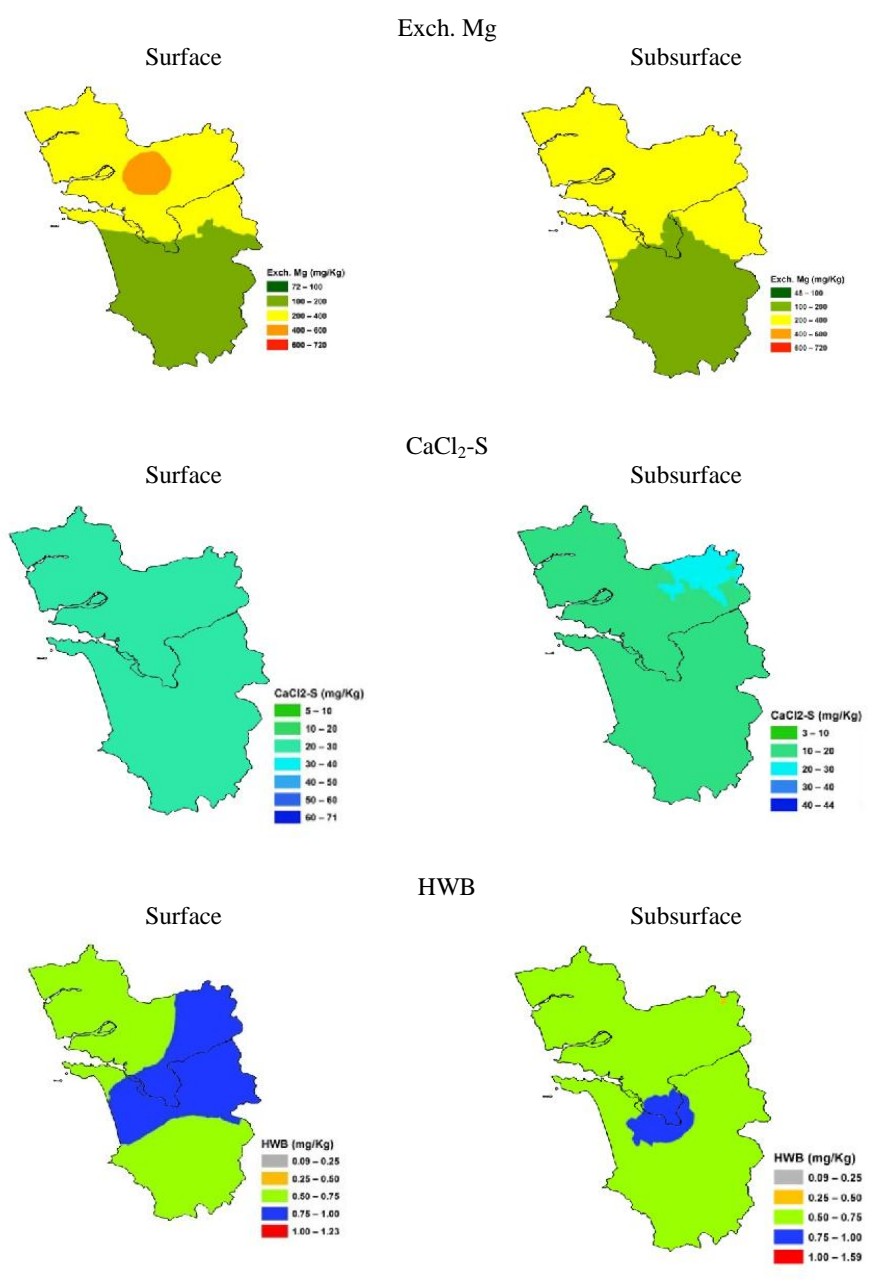

**Figure 2.** Kriged interpolation maps of soil properties in surface (0-20 cm) and subsurface (20-40 cm) soil layers