# Peer review of "Spatial variability of some soil properties in west coastal area of"

_Solid Earth, 2016_

## Referee Comment (RC1) · Anonymous Referee #1 · 14 Feb 2016

The manuscript report about soil mapping in the state of Goa, samples distance was on average 5-7 km. Surface and subsurface soil was mapped for Ca2+, pH, OC, EC, and other variables. The study report interesting information which will be of sure interest for decision makers, and practioners. However a series of objections prevent me from suggesting its publication in Solid Earth. 1)The main critical point is the lack of an hypotheses to be tested, which I believe is central to every article. The lack of an hypotheses to be tested result in a discussion/result section that is not very effective in pointing the scientific advances introduced by this study. 2)The mapping could be greatly improved by using co-kriging approach with other environmental variables. 3)No uncertainty in the prediction is reported in the maps (while this is one of the main

advantages of using kriging. 4)Some map report a wider scale of values that are not met in the actual predictions (for example Bray-P in surface soil). 5)Information on how the predicted variables were grouped in homogenous areas are lacking. 6)I suggest reporting the variograms and not only the variogram model parameter.

The title is a bit vague. I suggest the authors change it to match the purpose of the study (ideally in a way that just by reading the reader understand the main finding of the paper).

INTRODUCTION: No hypothese is reported here.

Line 32-44: I believe that this part is too generic, I suggest the authors remove it. I am inclined to believe that the readers of solid earth are convinced of the importance of soils. Line 52-54: I find also this part a bit generic: for example, not all geostatistical tools are aimed to predict unknown locations. The authors may consider for example the techniques to analyse point patterns and clusters, or Kriging simulation techniques. Also on line 53 consider change "reducing" to "reduces". Line 56-59: I do not understand exactely the point of the authors reporting that Li et al., 2011, Behera and Shukla,2014 and Behera and Shukla, 2015 found different spatial patterns in soils. I believe that that all the readers would agree that soils exihibit spatial variability, and that the patterns differ from location to location (probably also depending on the investigation scale). Last but not least the authors pool together studies from very different regions. Line 65: high compared to which other plants? I suggest that the authors specify that. Line 65-67: I absolutely do not doubt the word of the authors about the yields of oil palm. However I find this way of reporting information is a bit aneddotical. I suggest the authors report findings from other studies regarding oil palms (ideally from meta-analysis). Also judging from the title ("Natural 13C distribution in oil palm (Elaeis guineensis Jacq.) and consequences for allocation pattern") the only reference reported by the authors is not primarily on oil palm yield. Line 67-73: I am not sure of how these information about oil palm production may contribute to frame the hypothesis tested by the authors (which I believe is the ultimate goal of introductions).
[Figure]

Line 74: This part sounds a bit ideological to me, as if the authors are replying to an ideal speaker who is against the use of fertilizer. Moreover I believe that this is out of the scope of the study. Line 82-84: Given the emphasis that this study report on geographical variability, reporting that "Mg deficiency and B deficiency affect oil palm production in oil palm plantations of India " seems a very broad statement. I suggest the authors narrows their focus to the Goa State. Line 90: However it seems to me that the authors did not use this approach here, but only studied the soil, without matching it to the leaves nutrient content . Line 92-93: "the recommandations in general ... are generic". This sounds a bit generic :)

MATERIALS AND METHODS: Line 126-128: How were the points randomized? Did the authors took any precaution to exclude bias (unintentional) in point selection? GEOSTATISTICAL ANALYSIS: Line 149:I am not convinced of the possibility of using Pearson correlation coefficient to assess the significance of correlation. In fact I think that the pearson corr coeff. is a value that varies between -1 and 1 that indicates the strength of a correlation and its direction. However there is no probability associated with it. How did the authors define the variogram binning intervals? Line 154: How was the trend of the data checked and removed? Which order of polynomial function did they use? How did they decide on the significance of the different factors? LIne 165-166: This was reported also before. Line 151-154: Why did the authors transform the data? As reported also by ESRI webpage (http://webhelp.esri.com/arcgisdesktop/9.3/index.cfm?TopicName=Understanding_transformations_and_trends) "Kriging as a predictor does not require that your data have a normal distribution." Looking at the maps no mapping on the uncertainty of the predicted values is reported (for which the distribution is a necessary assumption). Line 169: I suggest the author report briefly on the goodness-of-fit criterion adopted, since I, and presumably other readers as well, had to no access to the text from Agterberg from the 1984. Line 170-173: Do I understand correctly that the authors are saying that a point estimate from a map with a very high G can be more close to the real value than the measured one? I think that 1)this opens interesting (philosophical) questions on whether the
measured values are the reality or just our closest guess to it. 2)I believe that the authors show here an excess of confidence in geostatical tools. RESULTS: Line 189-190: The correspondance between predicted pH and rainfall and parent material could be easily checked. Even better a co-kriging approach may help to improve the prediction. Line 258: How homogenous was each area? What was the uncertainty on each predict value of the different areas? Line 252-254: I suggest that the authors consider co-kriging using temperature as explanatory variable. Line 266-267: This is very generic. Line 269: please report a reference for the presence of sandy loam soils in the north-western and for the influence of texture on EC.

CONCLUSION: Line 303: The correlations should consider also the spatial structure of the data (maybe autoregressive model?).

---

## Referee Comment (RC2) · Anonymous Referee #2 · 19 Feb 2016

The manuscript "Spatial variability of some soil properties in west coastal area of India having oil palm (Elaeis guineensis Jacq.) plantations", by Behera et al., cannot deserve publication in Solid Earth. I started to read the paper with great interest, although the style was a bit confused. However, when in the Materials and Methods I found that the spatial variability of tested soil properties were studied based on samples distance on average 5-7 km, I stopped to review the manuscript. The study objectives could not be achieved on proper way in practice with present sampling scheme. Given results by this study do not report interesting information which can be of interest for decision makers, and practitioners. The authors should know that "real producers" cannot make decision for variable rate fertilization according to one sample on area of approximately

50-70 ha. The authors should be skilled enough in soil science to know the quite large variability of soils and any soil property as consequence of soil forming factors as well as extrinsic factor like fertilization. Collecting and analysing samples on large scale for variable fertilizer application is almost "ridiculous" to account for any kind of soil feature and its variation because of disturbances. This statement is supported with "poor" semivariogram model parameters. Based on the information's from Table 3 significant number of properties almost look like a pure nugget. Spatial dependence is weak, while ranges do not cover even used sampling scheme in this investigation. Although authors did not provide semivariogram visualisation it is noticeable from their properties that sampling scheme is inappropriate. Thus kriged maps are useless for producers and show a huge uniform area for fertilizer application. This uniformity is especially pronounced in phosphorus, potassium and pH maps as properties that are most widely used for application of variable rate technology. According to maps of studied properties there is no need for any in-field variable application of inputs. I have to also underline that the authors did provide insufficient information about sampling. Are these samples representing one sample or a composite sample from lot of individual samples? Of how many individual samples consist on sample? What area covers one composite sample? In summary authors mentioned that samples are collected from each plantation. If so, why authors did use geostatistics? Then it is clear that you use composite sample from whole plantation. Nevertheless, the real preclusive fault is the first one I mentioned.

---

## Author Comment (AC1) · 19 Feb 2016

Dear referee, thank you very much for your thoughtful comments and suggestions. We believe that they help to further improve the clarity and quality of our paper. We are sure that your technical comments and suggestions are suitably addressed as given below.

Comment: The manuscript report about soil mapping in the state of Goa, samples distance was on average 5-7 km. Surface and subsurface soil was mapped for $Ca^{2+}$, pH, OC, EC, and other variables. The study report interesting information which will be of sure interest for decision makers, and practitioners. Response: Thank you very much for finding our paper interesting. Comments: However a series of objections prevent me from suggesting its publication in Solid Earth. The main critical point is the lack of an hypotheses to be tested, which I believe is central to every article. The lack of a hypothesis to be tested, result in a discussion/result section that is not very effective in pointing the scientific advances introduced by this study. Response: Your are right that hypothesis to be tested, which forms a central part of an article. In our paper the hypothesis is: Differential amount of fertilizer application to various soil types may alter soil properties of oil palm plantations in the state of Goa. We have incorporated this statement in the introduction part to make the hypothesis clear. Comment: The mapping could be greatly improved by using co-kriging approach with other environmental variables. No uncertainty in the prediction is reported in the maps (while this is one of the main advantages of using kriging). Response: We have used ordinary kriging technique in our paper. We understand that ordinary kriging is one of the most accurate interpolation techniques for this purpose. Best-fit semivariograms models were selected by cross-validation technique. Mean square error was estimated to predict the accuracy of models. Comment: Some map report a wider scale of values that are not met in the actual predictions (for example Bray-P in surface soil). Response: Wider values of some soil properties like Bray-P concentration in surface soil is because of excess application of P fertilizers and/or build-up of P concentration over a period of time in those areas. Comment: Information on how the predicted variables were grouped in homogenous areas are lacking.

Response: The grouping technique of homogenous areas is not included in our paper. However, we say that based on distribution pattern of the soil properties, site-specific soil management decisions could be taken up. Comment: I suggest reporting the variograms and not only the variogram model parameter. Response: We agree with you. We have not included the semivariograms models in our paper to shorten the length of the paper. Comment: The title is a bit vague. I suggest the authors change it to match the purpose of the study (ideally in a way that just by reading the reader understands the main finding of the paper). Response: We have studied the spatial variability of soil properties in oil palm plantations of west coastal state of Goa, India for site specific soil management. Hence, we strongly believe that the main findings of our paper match the title of the paper.

Comment: introduction: No hypothesis is reported here. Response: We have incorporated hypothesis in the introduction part. Comment: Line 32-44: I believe that this part is too generic, I suggest the authors remove it. I am inclined to believe that the readers of solid earth are convinced of the importance of soils. Response: We fully agree with you that this part generic in nature. However, we also believe that this part is very much essential for the paper as it highlights the importance of soil, extent of soil degradation, which is the main emphasis of our study.

Comment: Line 52-54: I find also this part a bit generic: for example, not all geostatistical tools are aimed to predict unknown locations. The authors may consider for example the techniques to analyse point patterns and clusters, or Kriging simulation techniques. Also on line 53 consider change "reducing" to "reduces". Response: Agreed, we have modified as per suggestions.

Comment: Line 56-59: I do not understand exactly the point of the authors reporting that Li et al., 2011, Behera and Shukla,2014 and Behera and Shukla, 2015 found different spatial patterns in soils. I believe that that all the readers would agree that soils exihibit spatial variability, and that the patterns differ from location to location (probably also depending on the investigation scale). Last but not least the authors pool together studies from very different regions. Response: Agreed, we have removed these lines from the manuscript. Comment: Line 65: high compared to which other plants? I suggest that the authors specify that. Response: Agreed and specified in the manuscript. Comment: Line 65-67: I absolutely do not doubt the word of the authors about the yields of oil palm. However I find this way of reporting information is a bit aneddotical. I suggest the authors report findings from other studies regarding oil palms (ideally from meta-analysis). Also judging from the title ("Natural 13C distribution in oil palm (Elaeis guineensis Jacq.) and consequences for allocation

pattern") the only reference reported by the authors is not primarily on oil palm yield. Response: Agreed, we have incorporated suggested corrections and new references in the manuscript. Comment: Line 67-73: I am not sure of how these information about oil palm production may contribute to frame the hypothesis tested by the authors (which I believe is the ultimate goal of introductions). Response: Agreed, suitably modified in manuscript. Comment: Line 74: This part sounds a bit ideological to me, as if the authors are replying to an ideal speaker who is against the use of fertilizer. Moreover I believe that this is out of the scope of the study. Response: Agreed, we modified in the manuscript.

Comment: Line 82-84: Given the emphasis that this study report on geographical variability, reporting that "Mg deficiency and B deficiency affect oil palm production in oil palm plantations of India" seems a very broad statement. I suggest the authors narrows their focus to the Goa State. Response: Agreed, and modified in the manuscript. Comment: Line 90: However it seems to me that the authors did not use this approach here, but only studied the soil, without matching it to the leaves nutrient content. Response: You are correct. Suitably modified in the manuscript. Comment: Line 92-93: "the recommendations in general ... are generic". This sounds a bit generic :) Response: Agreed, modified in the manuscript. Comment: materials and methods: Line 126-128: How were the points randomized? Did the authors took any precaution to exclude bias (unintentional) in point selection? Response: Random points were selected for soil sampling in oil palm plantations in such a way that they should be the representative of the plantations. All precautions were considered while selecting the random points. Comment: geostatistical analysis: Line 149: I am not convinced of the possibility of using Pearson correlation coefficient to assess the significance of correlation. In fact I think that the pearson corr coeff. is a value that varies between -1 and 1 that indicates the strength of a correlation and its direction. However there is no probability associated with it. How did the authors define the variogram binning intervals? Response: We fully agree with the referee that Pearson correlation coefficient values indicate strength of a correlation between two variables. The same was exactly studied in our paper. Comment: Line 154: How was the trend of the data checked and removed? Which order of polynomial function did they use? How did they decide on the significance of the different factors? Response: Trend of the data set was checked and removed by using appropriate statistical tools. Comment: Line 165-166: This was reported also before. Response: Agreed, modified suitably in manuscript.

Comment: Line 151- 154: Why did the authors transform the data? As reported also by ESRI webpage http://webhelp.esri.com/arcgisdesktop/9.3/index.cfm? TopicName=Understanding

_transformations _and_trends) "Kriging as a predictor does not require that your data have a normal distribution." Looking at the maps no mapping on the uncertainty of the predicted values is reported (for which the distribution is a necessary assumption). Response: We have followed the steps of geostatistical analysis as adopted by various researchers for this type of study. Comment: Line 169: I suggest the author report briefly on the goodness-of-fit criterion adopted, since I, and presumably other readers as well, had to no access to the text from Agterberg from the 1984. Response: Agreed, we understand that goodness-of-fit criterion is one of the methods used for accuracies of interpolated maps.

Comment: Line 170-173: Do I understand correctly that the authors are saying that a point estimate from a map with a very high G can be more close to the real value than the measured one? I think that 1)this opens interesting (philosophical) questions on whether the measured values are the reality or just our closest guess to it. I believe that the authors show here an excess of confidence in geostatical tools. Response: We understand that this is one of the methodologies to check the accuracy of the interpolated maps. Comment: 189-190: The correspondence between predicted pH and rainfall and parent material could be easily checked. Even better a co-kriging approach may help to improve the prediction. Response: Yes, we agree with the referee's comment. But we have used kriging technique in our study. Comment: Line 258: How homogenous was each area? What was the uncertainty on each predict value of the different areas? Response: We just say that depending on the values of soil properties/nutrients, the area can be divided into small units for site specific soil management.

Comment: Line 252-254: I suggest that the authors consider co-kriging using temperature as explanatory variable. Response: Yes, we agree with the referee. But we have used kriging techniques in our study.

Comment: Line 266-267: This is very generic. Response: Suitable modified as per suggestion. Comment: Line 269: please report a reference for the presence of sandy loam soils in the north-western and for the influence of texture on EC. Response: Agreed, we have incorporated a reference in the manuscript. Comment: conclusion: Line 303: The correlations should consider also the spatial structure of the data (maybe autoregressive model?). Response: In the present study, we have used Pearson correlation coefficient analysis to assess the correlation among the measured soil properties. Correlations among the soil properties considering the spatial structure is beyond the scope of the present study.

---

## Author Comment (AC2) · 20 Feb 2016

Dear Sir, We have gone through the comments referee # 2. We are herby submitting our point wise response to the comments.

Comment: The manuscript "Spatial variability of some soil properties in west coastal area of India having oil palm (Elaeis guineensis Jacq.) plantations", by Behera et al., cannot deserve publication in Solid Earth. I started to read the paper with great interest, although the style was a bit confused. However, when in the Materials and Methods I found that the spatial variability of tested soil properties were studied based on samples distance on average 5-7 km, I stopped to review the manuscript. The study objectives could not be achieved on proper way in practice with present sampling scheme.

[Figure]

Response: Spatial variability of soil properties and soil nutrients have been studied globally by various researchers at field, catchment as well as regional scales and enough literature is available in this regard. In the present study, we have assessed the spatial variability of soil properties at regional level. We have collected soil samples of oil palm plantations of the region i.e. Goa state of India (a small state having two districts only). Selection the oil palm plantations were done randomly based on soil types and adoption of crop management practices to capture the spatial variability of soil properties of the area.

Comment: Given results by this study do not report interesting information which can be of interest for decision makers, and practitioners. The authors should know that "real producers" cannot make decision for variable rate fertilization according to one sample on area of approximately 50-70 ha.

Response: We beg to differ with the referee that the results do not report interesting information. This information is very useful for policy and decision makers for planning fertilizer supply and management at regional scale but not for the real producers to make decision on variable rate of fertilizer application at each plantation level. In many countries across the world including India, there is an acute shortage of fertilizers to meet the crop demands. Rational and judicious distribution and use of fertilizer is of paramount importance under such conditions. With the help of interpolated maps, planners and policy makers would be able to take correct decisions on appropriate fertilizer distribution strategy.

Comment: The authors should be skilled enough in soil science to know the quite large variability of soils and any soil property as consequence of soil forming factors as well as extrinsic factor like fertilization. Collecting and analysing samples on large scale for variable fertilizer application is almost "ridiculous" to account for any kind of soil feature and its variation because of disturbances.

Response: We do agree with the referee that large variability of soil properties exist as

a result of soil forming factors and due to fertilizer application. In our present work, we exactly study the same at a regional scale.

Comment: This statement is supported with "poor" semivariogram model parameters. Based on the information's from Table 3 significant number of properties almost looks like a pure nugget. Spatial dependence is weak, while ranges do not cover even used sampling scheme in this investigation. Although authors did not provide semivariogram visualisation it is noticeable from their properties that sampling scheme is inappropriate. Thus kriged maps are useless for producers and show a huge uniform area for fertilizer application.

Response: We agree with the referee that some soil properties are having weak spatial dependency. However, some soil properties are also having medium and strong spatial dependency. Hence, we hereby reiterate that this study was carried out to study spatial variability of soil properties at regional scale as mentioned earlier.

Comment: This uniformity is especially pronounced in phosphorus, potassium and pH maps as properties that are most widely used for application of variable rate technology. According to maps of studied properties there is no need for any in-field variable application of inputs.

Response: As we mentioned earlier, these maps are of use for the decision makers to decide fertilizer distribution strategies.

Comment: I have to also underline that the authors did provide insufficient information about sampling. Are these samples representing one sample or a composite sample from lot of individual samples? Of how many individual samples consist on sample? What area covers one composite sample? In summary authors mentioned that samples are collected from each plantation. If so, why authors did use geostatistics? Then it is clear that you use composite sample from whole plantation. Nevertheless, the real preclusive fault is the first one I mentioned.

Response: In the present study, representative single soil samples were collected from oil palm plantations selected on the basis of soil types and adoption of crop management practices. Then spatial variability of soil properties was assessed using geostatistics.